

# Construction and validation of a novel prognostic signature of microRNAs in lung adenocarcinoma

Wanzhen Li[1,*], Shiqing Liu[2,3,4,*], Shihong Su[1], Yang Chen[1] and Gengyun Sun[1]

[1] Department of Respiratory and Critical Care Medicine, The First Affiliated Hospital of Anhui Medical University, Hefei, China
[2] Department of Respiratory Medicine, National Key Clinical Specialty, Branch of National Clinical Research Center for Respiratory Disease, Xiangya Hospital, Central South University, Changsha, China
[3] Key cite of National Clinical Research Center for Respiratory Disease, Xiangya Hospital, Central South University, Changsha, China
[4] Xiangya Lung Cancer Center, Xiangya Hospital, Central South University, Changsha, China
* These authors contributed equally to this work.

Corresponding authors
Shiqing Liu, 451063788@qq.com
Gengyun Sun, sungengy@126.com

## ABSTRACT

MicroRNA (miRNA, miR) has been reported to be highly implicated in a wide range of biological processes in lung cancer (LC), and identification of differentially expressed miRNAs between normal and LC samples has been widely used in the discovery of prognostic factors for overall survival (OS) and response to therapy. The present study was designed to develop and evaluate a miRNA-based signature with prognostic value for the OS of lung adenocarcinoma (LUAD), a common histologic subtype of LC. In brief, the miRNA expression profiles and clinicopathological factors of 499 LUAD patients were collected from The Cancer Genome Atlas (TCGA) database. Kaplan–Meier (K-M) survival analysis showed significant correlations between differentially expressed miRNAs and LUAD survival outcomes. Afterward, 1,000 resample LUAD training matrices based on the training set was applied to identify the potential prognostic miRNAs. The least absolute shrinkage and selection operator (LASSO) cox regression analysis was used to constructed a six-miRNA based prognostic signature for LUAD patients. Samples with different risk scores displayed distinct OS in K-M analysis, indicating considerable predictive accuracy of this signature in both training and validation sets. Furthermore, time-dependent receiver operating characteristic (ROC) analysis demonstrated the nomogram achieved higher predictive accuracy than any other clinical variables after incorporating the clinical information (age, sex, stage, and recurrence). In the stratification analysis, the prognostic value of this classifier in LUAD patients was validated to be independent of other clinicopathological variables, such as age, gender, tumor recurrence, and early stage. Gene set annotation analyses were also conducted through the Hallmark gene set and Kyoto Encyclopedia of Genes and Genomes (KEGG) pathways, indicating target genes of the six miRNAs were positively related to various molecular pathways of cancer, such as hallmark UV response, Wnt signaling pathway and mTOR signaling pathway. In addition, fresh cancer tissue samples and matched adjacent tissue samples from 12 LUAD patients were collected to verify the expression of miR-582's target genes in the model, further revealing the potential relationship between SOX9, RASA1, CEP55, MAP4K4 and LUAD tumorigenesis, and validating the predictive value of the model. Taken together, the present study identified a robust signature for the OS prediction of LUAD patients,

which could potentially aid in the individualized selection of therapeutic approaches for LUAD patients.

## INTRODUCTION

Lung cancer (LC) is the leading cause of malignancy-related mortality, with approximately one million cases of death annually, which is a great burden for public health worldwide (*Torre et al., 2015*). Lung adenocarcinoma (LUAD) is the most common histologic subtype of LC (*Brambilla et al., 2001*; *Wakelee et al., 2007*), accounting for 50% of LC cases. Despite the remarkable progress in image scan, bronchoscopy, and novel therapeutic options (*Sun, Schiller & Gazdar, 2007*; *Minna, Roth & Gazdar, 2002*), tumor progression remains the major obstacle for a favorable survival rate. Additionally, 40% of stage IB and 66% of stage II patients are facing a high recurrence rate within 5 years, which is associated with a poor prognosis (*Chansky et al., 2010*). Recent studies have shown that adjuvant chemotherapy confers a 4–15% survival benefit for stage II-III patients receiving tumor resection rather than patients with stage I (*Reck et al., 2013*; *Pignon et al., 2008*). The limited survival benefits suggest the insufficient power of the currently available staging system and the presence of more undiscovered tumor heterogeneity. Therefore, it is imperative to identify early diagnostic markers and establish effective predictive systems for prognosis to improve the survival of LUAD patients.

MicroRNA (miRNA, miR) is a key component of a non-coding RNA family that regulates gene expression at a post-transcriptional level (*Garzon, Calin & Croce, 2009*). By specifically combing to the 3′untranslated region (UTR) of target genes and affecting messenger RNA (mRNA), miRNA has been proved to be involved in the regulation of basic cellular processes, including cell differentiation, proliferation, and apoptosis (*Winter et al., 2009*). Differentially expression levels of miRNAs, which may exhibit either oncogenic or tumor-suppressive activity, are reported to play crucial roles in the prognosis of LC (*Li et al., 2019a*; *Li et al., 2019b*; *Liu et al., 2018*; *Garzon, Calin & Croce, 2009*; *Lujambio et al., 2007*). The role of miRNA in lung carcinogenesis was first proposed in 2004 by *Calin et al. (2004)*, who demonstrated that half of the miRNA genes were located at cancer-related genomic regions or in fragile sites, and several miRNAs in these regions had low expression level in LC cell lines. Consistently, the down-regulation of the let-7 miRNA was found by another study published in the same year (*Takamizawa et al., 2004*), and acted as onco-suppressor role in LUAD carcinogenesis (*Kumar et al., 2008*). Junichi et al. have shown that down-regulation of the let-7 miRNA could prohibit the expression of RAS, MYC, and HMGA2 oncogenes (*Lee & Dutta, 2007*; *Kumar et al., 2007*). While *Johnson et al. (2005)* have reported that the over-expression of let-7 in a LC cell line (A549) could cause cell cycle arrest and inhibit cell growth. The miR-34 family, consisting of miR-34a, miR-34b, and miR-34c members, has long been reported to be involved in the p53 regulatory network

(*He et al., 2007*). By directly targeting MET and BCL2, the downregulation of miR-34 could promote cell growth and proliferation (*Bommer et al., 2007*). Besides, a novel tumor evasion mechanism has been proposed, indicating that PDL1 was regulated by p53 via miR-34 in PD1/PDL1 signaling in non-small cell lung cancer (NSCLC) patients (*Cortez et al., 2016*).

Currently, miRNAs have been increasingly used not only as diagnostic but also prognostic biomarkers due to the convenient detection in body fluids such as blood, serum, urine, and sputum. Several studies have suggested the existence of tissue-specific miRNA signatures, which might be used to classify different cancer types (*Zhang et al., 2019a*; *Zhang et al., 2019b*; *Xiong et al., 2018*; *DeSantis et al., 2014*; *Chansky et al., 2010*). Therefore, it is promising to find out some prognosis-related miRNA markers to establish the prognostic prediction signature. Herein, we established an overall survival (OS) predictive signature based on a six-miRNA-based panel, followed by validation of its usage and stability in prognostic prediction in LUAD patients.

## MATERIAL AND METHODS

### Data and sources

The preprocessed LUAD mature miRNA expression profiles in the Cancer Genome Atlas (TCGA) database displayed as log2 converted reads per million (log2(RPM + 1)) were downloaded, along with the clinical information of related samples including survival follow-up data, age, gender, tumor node metastasis (TNM) stage, and tumor recurrence. The miRNA expression data from the LUAD samples were based on IlluminaHiSeq_miRNASeq, followed by quantile normalization and log2-scale transformation by R software. All the enrolled samples were 2:1 random sampling without replacement by using sample R package. The randomizing process was blind to the demographic information or clinical information, which could ensure that the training and validation cohorts are independent to clarify the key points we focused on. We use R (http://www.R-project.org) software to make all statistical analyses. A two-sided $P < 0.05$ was considered statistically significant.

Herein, the predictive model was initially constructed in the training set, and the validation group was employed to provide the unbiased evaluation of the final model in the training set.

In addition, 12 cases of fresh LUAD tissue samples and paired adjacent cancer tissue samples were enrolled from the Department of Thoracic Surgery, the First Affiliated Hospital of Anhui Medical University. All the 24 surgically resected samples were immersed in 10% formalin. Informed consents were signed by all patients and our study was approved by the Ethics Committee of the First Affiliated Hospital of Anhui Medical University (anyiyifuyuanlunshen-Quik-PJ-2019-09-09) (anyiyifuyuanlunshen-Quik-PJ-2020-05-08). For the patients collected from the TCGA project, the ethics and policies were announced on their Website (https://www.cancer.gov/about-nci/organization/ccg/research/structural-genomics/tcga/history/policies).

## Establishment and validation of a miRNA signature with the least absolute shrinkage and selection operator (LASSO) regression model

The prognostic significance of every single miRNA was preliminarily assessed by Kaplan–Meier (K-M) survival analysis, followed by a log-rank test to select all the potential miRNAs. The candidate miRNA was considered to be significantly associated with the OS of LUAD patients when the *P*- value was less than 0.05 (*Xu et al., 2017*). A group of 1,000 training matrices based on the LUAD patients among the training set was subsequently obtained after the resample model inclusion probability (RMIP). Followed, with the R package "glmnet" (2.0–10), the LASSO bagging cox regression test, the classical and modified method in Cox regression analysis of high dimensional data was performed with 10-fold cross-validation on all the OS related matrices (*Tibshirani, 1997*). The tuning parameter λ was determined by the standard error (1-SE), which introduced the penalties to the model construction process to avoid over-fitting. The larger λ was, the more the coefficients were shrunk toward zero (*Goeman, 2010*). In this study, the recommended λ value with minimal mean squared error plus one standard deviation was selected in each Lasso cox regression test. A group of 1,000 matrices was generated after resampling the data points 1,000 times of the training group, and the LASSO Cox regression test was performed. A list of miRNAs, which displayed more than 500 bootstraps (2/3 of all) during the down-sampling test, was eventually picked up as prognostic biomarkers. Finally, an individual's risk score model for each patient was constructed for predicting prognosis of LUAD patients by including the expression level of each optimal prognostic miRNA, weighted by their estimated regression coefficients of LASSO Bagging Cox regression model as follows:

Risk Score (patient) $= \sum$ i coefficient miRNA Þexpression miRNAiðÞ.

The risk score of each LSCC patient was calculated according to the prognostic predicting formula. Afterward, the LUAD samples with assigned risk scores were classified into a high-risk or low-risk group by using the median as the cutoff point. To confirm the predictive ability of this model, K-M survival curves with the log-rank test in the training and validation sets were also conducted to distinguish the survival difference between the high-risk group and low-risk group. A two-tailed *P*-value < 0.05 in K-M analyses was considered as statistically significant. In addition, to graphically show the diagnostic value of the classifier system, receiver operating characteristic (ROC) analysis was adopted to compare the sensitivity and specificity of survival prediction. Youden's index = max [1-(sensitivity + specificity)] (*Hilden & Glasziou, 1996*). Moreover, the area under the ROC curve (AUC) for evaluating discriminatory ability was calculated as well.

## Network and annotation analysis of target genes

We obtained the target genes from TargetScan, miRTarBase, and miRDB (*Wong & Wang, 2015*; *Chou et al., 2018*; *Agarwal et al., 2015*), respectively. To obtain more convincing results, we overlapped the target genes from the three websites and generated a new target gene cluster. Cytoscape software (v3.5.1, San Diego, La Jolla, California 92093, USA) was used for the visualization of the interactive network to assist the understanding miRNA

function and regulatory mechanisms. Besides, Hallmark gene set and Kyoto Encyclopedia of Genes and Genomes (KEGG) analyses were performed to further study the functional roles of the target genes during LUAD progression using the R package "clusterProfiler" ($Q$-value < 0.05) (*Yu et al., 2012*).

### Immunohistochemistry (IHC)

We randomly chose the SOX9, RASA1, CEP55 and MAP4K4, the target genes of miR-582, for IHC validation, which was performed according to the previous description (*Zhang et al., 2019a*; *Zhang et al., 2019b*). The *t*-test was used to determine the protein expression differences between LUAD tissue specimens and paired adjacent cancer tissue specimens. A two-tailed *P*-value < 0.05 was considered as statistically significant. The SOX9, RASA1, CEP55 and MAP4K4 antibodies (catalog # GR3253329-3; GR85944-25; GR234851-12; 09000421) were purchased from Abcam (Cambridge, UK; http://www.abcam.com) and Proteintech (Chicago, IL, USA; http://www.ptglab.com).

## RESULTS

### Construction of a prognostic signature based on miRNAs

The miRNA expression profile and clinical information of 518 LUAD cases were downloaded from the TCGA databases. OS information was available in 499 cases, which were further retained for downstream analysis. Among LUAD patients, the samples were randomly divided into two groups as the training set (329 samples) and validation set (170 samples).

The K-M survival analysis was employed for the preliminary screening of potential OS related miRNAs (Fig. S1). As a result, we selected eight miRNAs based on $P < 0.05$, which indicated significant correlation with the LUAD patients' OS. Afterwards, we rebuilt 1,000 resample LUAD training matrices based on the training cohort patients. The RMIP of each OS-related miRNA was listed and ordered from the most frequent one to the least miRNA (Fig. 1A). According to the LASSO cox regression analysis on performance of all the OS related matrices with 10-fold cross validation, six miRNAs were acquired, including miR-1468 (HR = 0.69, 95% CI [0.560–0.870], $P = 0.001$, co-ef = −0.366), miR-299 (HR = 1.48, 95% CI [1.220–1.800], $P < 0.001$, co-ef = 0.394), miR-4709 (HR = 0.720, 95% CI [0.590–0.890], $P = 0.002$, co-ef = −0.322), miR-5571 (HR = 0.570, 95% CI [0.370–0.860], $P = 0.008$, co-ef = −0.570), miR-582 (HR = 1.18, 95% CI [1.050–1.320], $P = 0.005$, co-ef = 0.166), and miR-3653 (HR = 0.87, 95 % CI [0.730–1.030], $P = 0.106$, co-ef = −0.143) (Fig. 1B). A risk score model for each patient was constructed to predict prognosis of LUAD patients by including the expression level of each optimal prognostic miRNA, weighted by their estimated regression coefficients of LASSO Bagging Cox regression model as = −0.365 * miR-1468 + 0.394 * miR-299 -0.322 * miR-4709 −0.570 * miR-5571 + 0.166 * miR-582 - 0.143 * miR-3653.

### Validation and evaluation of the miRNAs signature

Using the median risk score as the cut-off value, the LUAD patients were assigned into high-risk or low-risk group. The K-M curve with the log-rank test demonstrated significantly

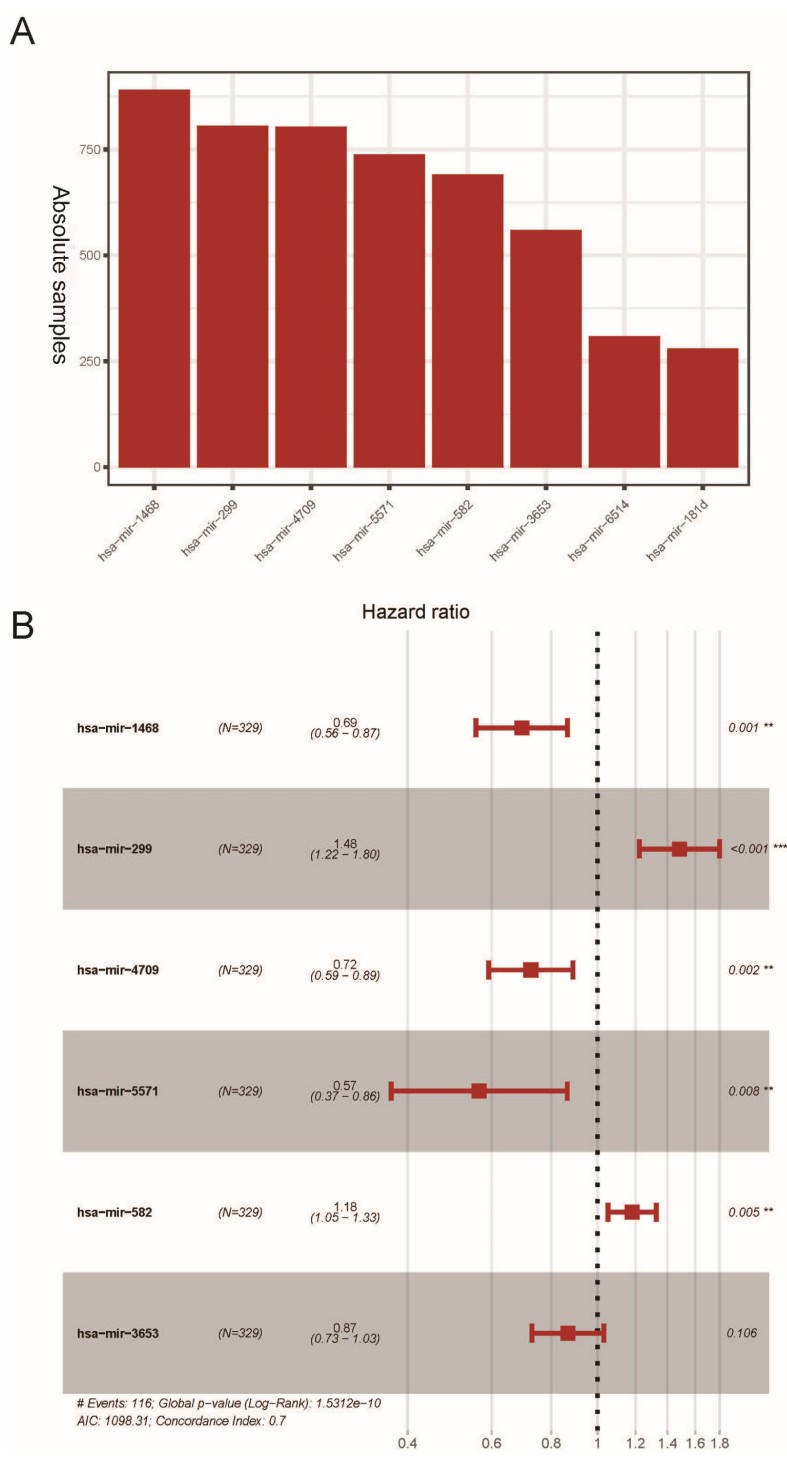

**Figure 1** **Construction of the six miRNAs predicted signature associated with OS of LUAD.** Distribution of 1,000 times resampled results in the top eight OS-related miRNA (A); the Hazard ratio of enrolled OS-related miRNA (B) conducted by LASSO Cox regression analysis. OS, overall survival; LASSO, the least absolute shrinkage and selection operator; LUAD, lung adenocarcinoma.

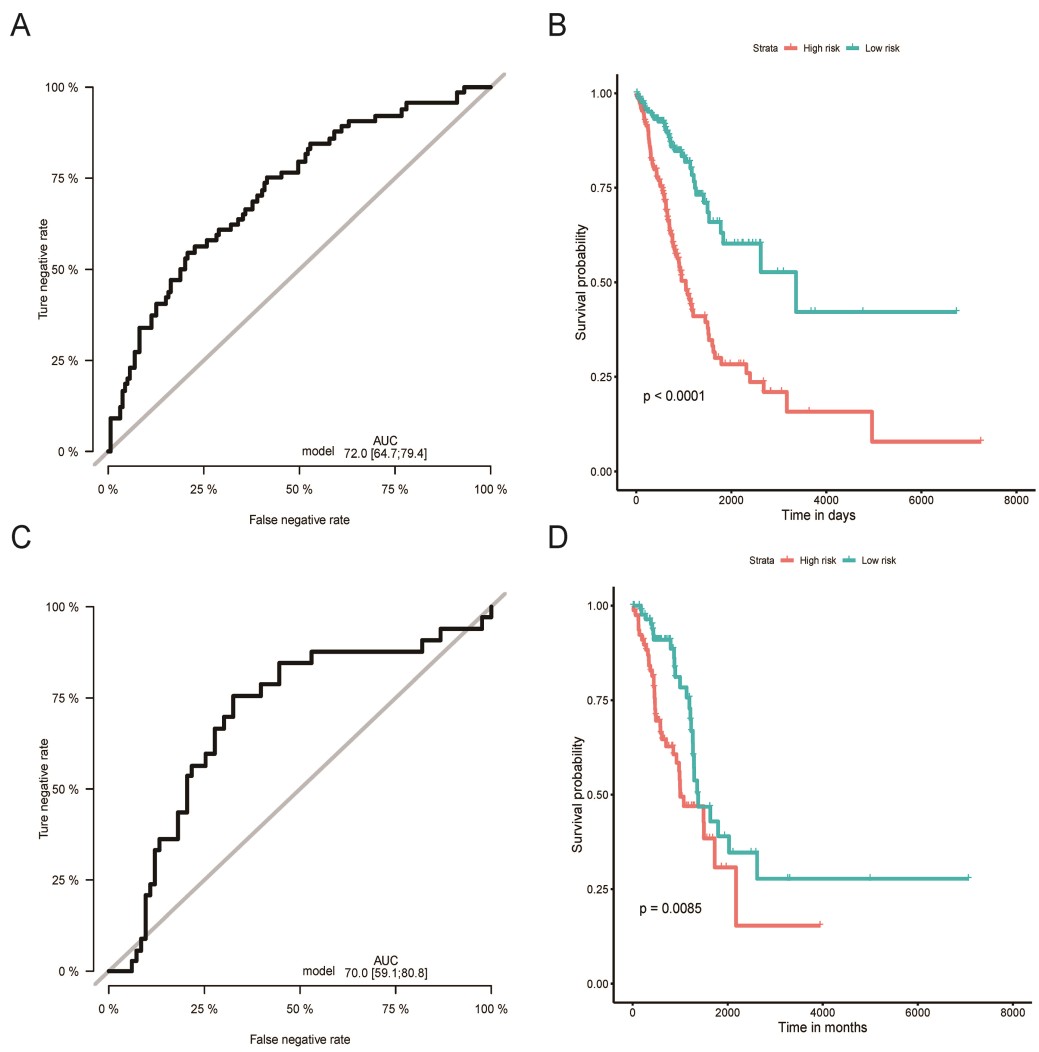

**Figure 2 Evaluation of OS miRNA signature predictive performance.** Kaplan–Meier curves of the low- and high-risk groups divided by six miRNA-based OS predictive signature in training cohort (A), and validation cohort (C); ROC curves of the low- and high-risk groups divided by six miRNA-based OS predictive in training cohort (B), and validation cohort (D). OS, overall survival; ROC, receiver operating characteristic.

different survival between high-risk and low-risk groups in both training ($P < 0.0001$, Fig. 2B) and validation groups ($P = 0.0085$, Fig. 2D). To evaluate the classification potential of this model, the ROC analysis was performed and AUC was calculated in both training and validation cohorts. As shown in Figs. 2A and 2C, the AUC of ROC was 0.72 and 0.70 in the training group and validation group, respectively. Besides, considering other partitions to training and validation groups can both increase and decrease the reliability of the classifier, we repeated the step of different training and validation groups multiple times and efficiently calculated the means/variations/confidence intervals for the sensitivity, specificity and AUC to validate the reliability of the classifier as the Table S3. These results proved that the classifier has moderate discriminatory power and potential

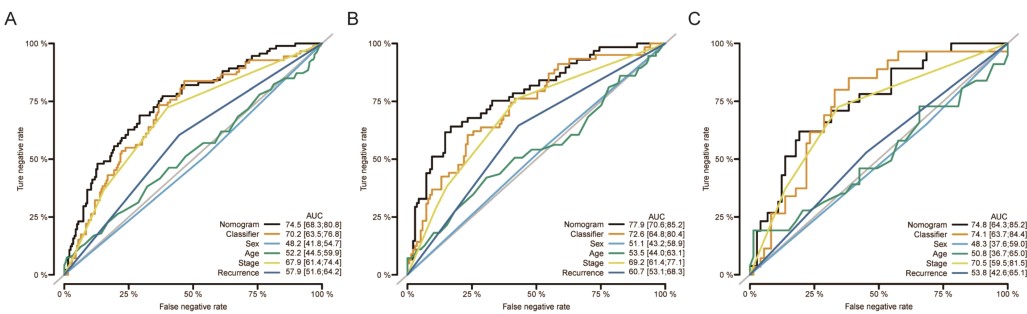

**Figure 3 The comparison between the six-miRNA-based OS classifier and clinicopathological features.** ROC curve showed the different performance of prognostic value between different variables in the overall set (A), training set (B), and validation set (C). Nomogram was a synthesis model by combining the miRNA-based overall survival classifier and clinicopathological features. AUC, the area under the curve; ROC, receiver operating characteristic. OS, overall survival.

utility in determining the LUAD patients with low or high risk as presenting high and stable of AUC and sensitivity.

Moreover, nomograms were constructed in the training, validation, and overall groups after incorporating the clinical characteristics (age, sex, TNM stage, and tumor recurrence) (Figs. 3A–3C). The results of ROC analyses identified this six-miRNAs classifier as the most accurate contributor to prognosis compared to other clinical factors, and the nomogram could precisely and steadily judge the OS rate of LUAD.

## Subgroup analysis

Age, TNM stage, and tumor recurrence are always important clinical factors influencing the prognosis of LUAD. Therefore, stratification analyses based on TNM stage, tumor recurrence, and other clinical variables were performed to validate whether this classifier had functional discriminatory capacity among different clinical status (Fig. 4). According to the TNM staging system, LUAD patients have categorized two groups (stage I/II, stage III/IV). Within Stage I/II, the OS related to the six miRNAs signature prognostic value remains significantly different (stage I/II: $P < 0.001$), but not in Stage III/IV subgroup (stage III/IV: $P = 0.061$). When adjusted by age, sex, and new tumor event after initial treatment, our risk score model still exhibited the reliability and general applicability for distinguishing each group among different clinical factors.

## Functional annotation analysis and IHC Validation

The target genes of miR-1468, miR-299, miR-4709, miR-5571, and miR-582 were predicted by the online prediction tool like TargetScan, miRTarBase, and miRDB. The miRNA-miRNA regulatory network was shown in Fig. 5. Additionally, we performed functional annotation analysis for Hallmark gene set and KEGG pathways (Fig. 6), demonstrating that target genes of different miRNAs were significantly enriched in several functional pathways related to cancer development, such as hallmark UV response, Wnt signaling pathway and mTOR signaling pathway.

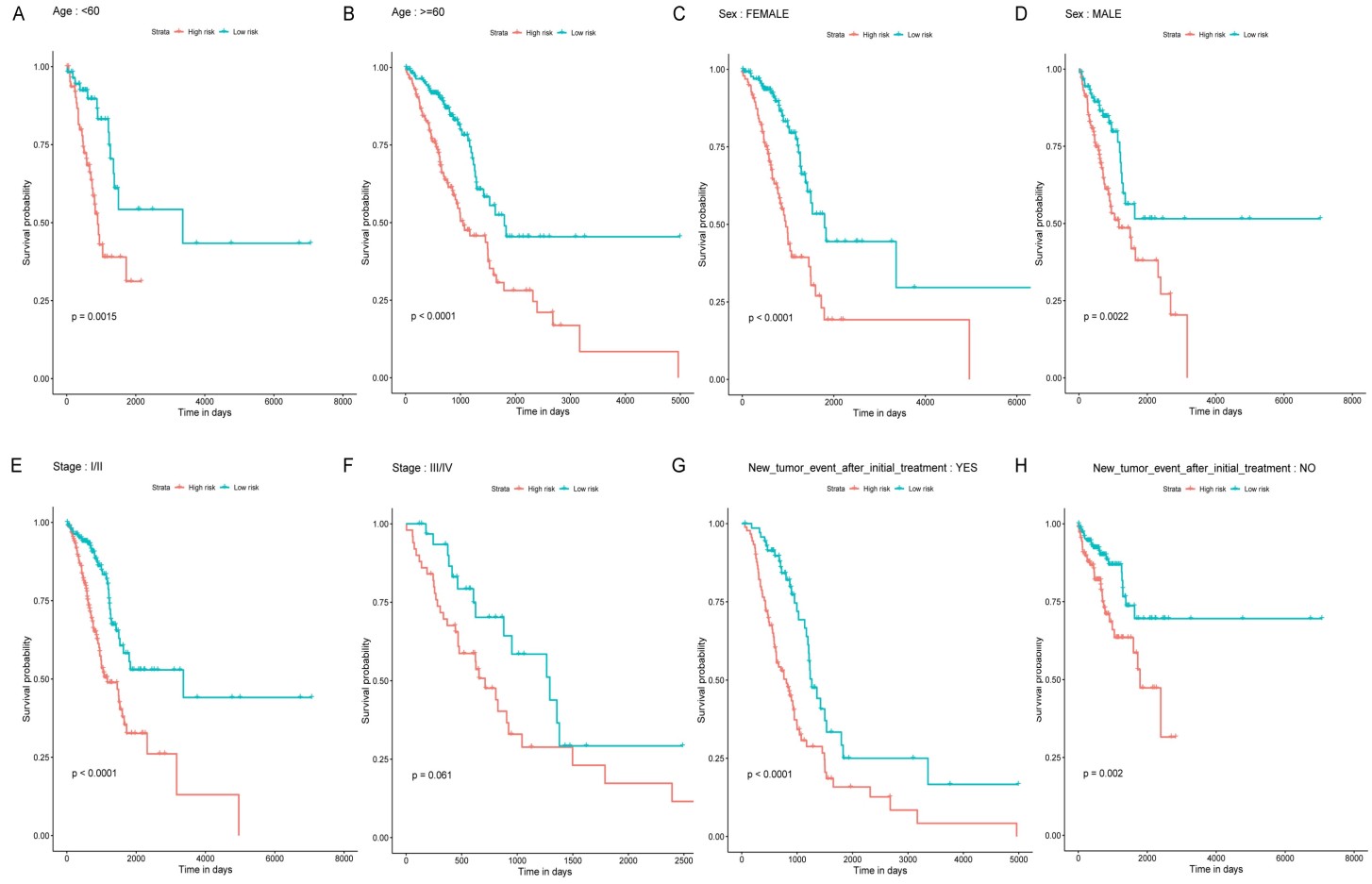

**Figure 4** **Stratified survival analyses of the LUAD patients according to the miRNA-based OS predicting classifier.** Kaplan-Meier survival analyses for the patients with different characteristics including age, sex, tumor stage, and new tumor event after initial treatment (A-H) according to the miRNA-based OS predicting classifier. LUAD, lung adenocarcinoma; OS, overall survival.

Moreover, to confirm the protein expression pattern of the target genes in the LUAD tissue, the expression of SOX9, RASA1, CEP55, and MAP4K4 was detected in LUAD clinical specimens by IHC. The results showed that the LUAD patients mostly had high SOX9, RASA1, CEP55 and MAP4K4 protein expression ($p < 0.05$) (Figs. S2–S5).

## DISCUSSION

Characterized by the tendency of an advanced stage and metastatic tumor, LUAD confronts worse survival outcomes than many other types of cancer, with a 5-year survival rate less than 18% (*Chansky et al., 2010*). Early-stage LUAD patients might theoretically achieve the highest survival outcomes, though it accounts for only 25%–30% of LC (*DeSantis et al., 2014*; *Scott et al., 2007*). Surgical resection with or without additional adjuvant chemotherapy is regarded as the cornerstone of therapeutic treatment for early-stage LUAD, and TNM stage is the most significant biomarker to predict prognosis traditionally (*Crinò et al., 2010*; *Vansteenkiste et al., 2014*; *Cheng et al., 2012*). Whereas, survival varies

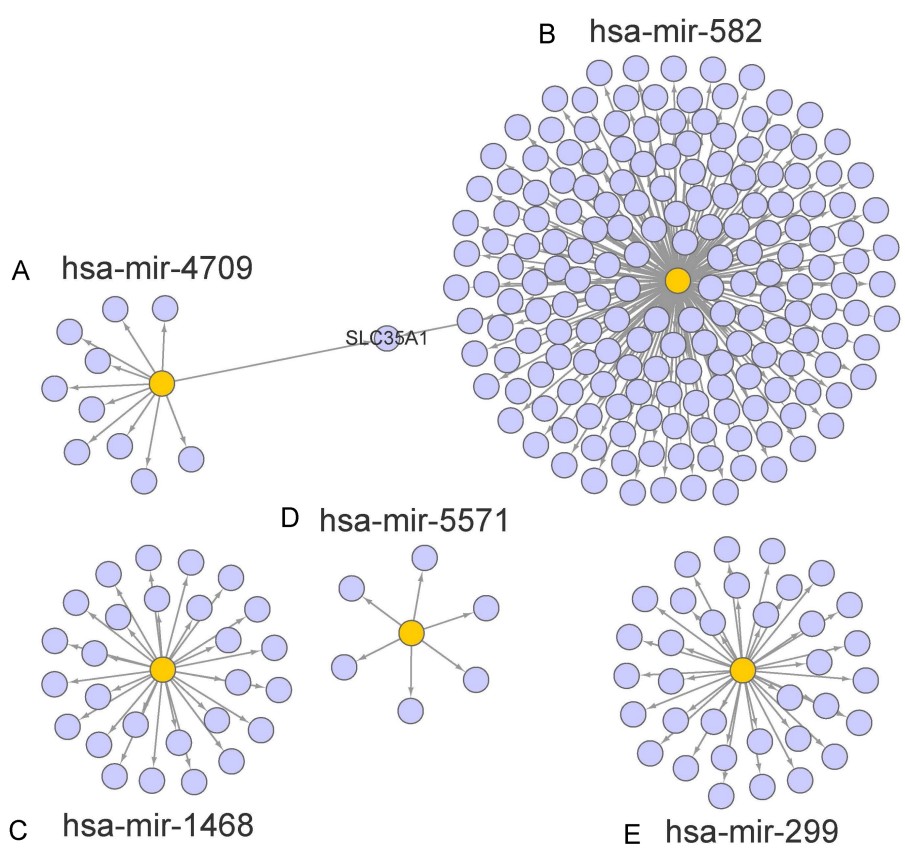

**Figure 5** **Prediction of miRNAs down-stream target genes.** We obtained the targeted genes of OS-related hsa-miR-4709, hsa-miR-582, hsa-miR-1468, hsa-miR-5571, hsa-miR-299 (A–E) from TargetScan, miRTarBase and miRDB, respectively, and overlapped the targeted genes to generated and visualized a new targeted gene cluster. OS, overall survival.

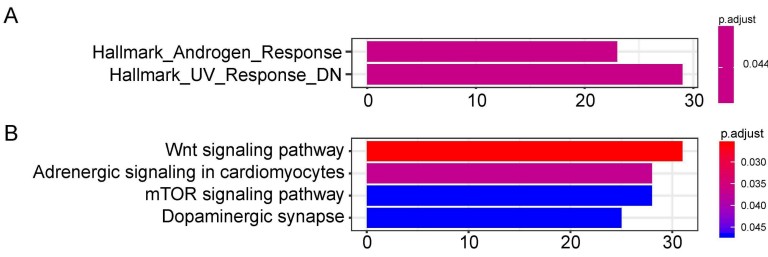

**Figure 6** **Functional enrichment analysis depicted the biological pathways and processes associated with OS correlated genes.** Hallmark enrichment (A) and KEGG signaling pathways analysis (B) were conducted to classify the functions of these overlapped target genes of OS related hsa-miR-1468, hsa-miR-299, hsa-miR-4709, hsa-miR-5571, hsa-miR-582, respectively. OS, overall survival.

greatly, even within the same-staged LUAD receiving the same treatment, demonstrating the genetic heterogeneity among LUAD.

In recent years, substantial cancer-related studies have focused on miRNAs profiles, as exceptionally influencing developmental and oncogenic pathways by regulating gene expression. Aberrant expression of miRNAs is considered as hallmark features in the biological progression of various diseases, including LC (*Ortholan et al., 2009*; *Saito et al., 2011*; *Patnaik et al., 2010*). Integration of miRNAs and target genes profiles is a promising approach to identify effective biomarkers of LUAD (*Zhu et al., 2019*; *Zhao et al., 2013*; *Jacob et al., 2018*). In our work, we obtained the miRNA expression profiles from TCGA databases and subsequently constructed a six-miRNAs based predictive signature of the prognosis for LUAD patients. This miRNAs-based signature could classify the high-risk group and low-risk group of LUAD with distinct difference survival. After incorporating the clinical parameters (age, sex, TNM stage, and tumor recurrence), the six-miRNA based nomogram exhibited favorable discriminative performance. In respect to stratification analyses based on clinicopathological variables, we validated the potential clinical utility of miRNA-based predictive signature for LUAD's OS, independent of age (>60, and <60), gender, tumor recurrence, and the early stage.

In consideration of the individual roles of six miRNAs (miR-1468, miR-299, miR-4709, miR-5571, miR-582, and miR-3653), several studies have reported their imperative involvements in prognosis value of LC as well. Niu et al. investigated that the mRNA expression of PIP4K2A regulated by hsa-miR-1468 was directly correlated with the OS of paclitaxel-treated LC (*Niu et al., 2012*). When the differentially expressed miRNAs were subjected to univariate Cox proportional hazard regression analysis, Lin et al. pointed out that miR-1468 ($P = 0.009$) and miR-3653 ($P = 0.012$) were significantly correlated with recurrence-free survival (RFS) of LUAD (*Lin et al., 2016*). The down-regulated expression of miR-299 in LUAD has been reported to serve as a functional role in therapeutic efficacy, including gefitinib resistance and chemoresistance (*Negrete-Garcia et al., 2018*; *Li et al., 2018*; *Zheng et al., 2015*). Through miR-299-3p/TNFSF12 pathway, RHPN1-AS1 could modulate the LUAD therapeutic resistance to gefitinib (*Li et al., 2018*), and the suppression of ABCE1 could affect the sensitivity of LC to doxorubicin (*Zheng et al., 2015*). Based on the detection in both tissues and plasma of LUAD patients with different stages, Pu et al. revealed that constantly expressed miR-5571-5p was a promising biomarker of LUAD with different stages (*Pu et al., 2016*), which was consistent with the study by Zheng et al. (*Zheng et al., 2018*). MiR-4709 and miR-582 in this signature were initially documented as prognostic markers in different tumors (*Siriwardhana et al., 2019*; *Xin et al., 2019*), which were validated by the microarray technology or TCGA. The miR-582 was further shown to be significantly down-regulated in NSLCLC cell lines, which could suppress the proliferation, migration, and invasion of NSLCLC cells by targeting MAP3K2 (*Wang & Zhang, 2018*).

We also presented the biological functions of these six miRNAs at the molecular level using the Hallmark gene set and KEGG analyses. As a result, the identified targets genes were mainly enriched in cancer-related pathways, such as hallmark UV response, Wnt signaling pathway, and mTOR signaling pathway. Wnt signaling pathways could significantly affect tumorigenesis, prognosis, and therapeutic resistance in NSCLC (*Stewart, 2014*). Activation of FOXP3 could facilitate the Wnt-β-catenin signaling pathway in epithelial–mesenchymal

transition (EMT), tumor growth, and metastasis of LUAD (*Yang et al., 2017*). MTOR signaling pathways were involved in regulating autophagy, apoptosis, proliferation, and metastasis in the pathophysiology of LUAD (*Liu et al., 2017*; *Han et al., 2013*; *Yu et al., 2017*). In order to assess the potential target genes of miRNAs, IHC was used to randomly identify the expression of SOX9, RASA1, CEP55 and MAP4K4, the target genes of miR-582. The results revealed that SOX9, RASA1, CEP55, and MAP4K4 were significantly higher expressed in LUAD tissues than those in adjacent cancer tissues. As wang et al. have validated that the expression of miR-582 was significantly down-regulated in NSLCLC cell lines and clinical specimens (*Wang & Zhang, 2018*), it is hypothesized that down-regulated miR-582 could inversely lead to the over-expression of its target genes, which is consistent with our results. However, the biological function of these target genes in LUAD has not been investigated in our study. In further research, we will conduct well-designed and comprehensive functional and mechanistic experiments.

In summary, we have identified six miRNAs related to the OS of LUAD and constructed a prognostic model. Survival analysis indicated that our risk score model was of significant prognostic value for LUAD patients, which had the potentiality to facilitate the decision-making of individualized treatment decisions for LUAD patients. By randomly assessing the potential target genes in the tumor samples, we have found that up-regulated SOX9, RASA1, CEP55 and MAP4K4, the target genes of miR-582, might play important roles in LUAD tumorigenesis. However, the underlying mechanism should be further investigated. Collectively, our model is promising for clinical prognostic evaluation of LUAD patients.

### Funding
This work was supported by the National Key Clinical Specialist Construction Programs of China (N3101005005025), the National Multidisciplinary Cooperative Diagnosis and Treatment Capacity Building Project for Major Diseases (Lung Cancer), and the Xiangya clinical big data project of Central South University (Clinical big data project of lung cancer). The funders had no role in study design, data collection and analysis, decision to publish, or preparation of the manuscript.

### Grant Disclosures
The following grant information was disclosed by the authors:
The National Key Clinical Specialist Construction Programs of China: N3101005005025.
The National Multidisciplinary Cooperative Diagnosis and Treatment Capacity Building Project for Major Diseases (Lung Cancer).
The Xiangya clinical big data project of Central South University (Clinical big data project of lung cancer).

### Competing Interests
The authors declare there are no competing interests.

## Author Contributions

- Wanzhen Li conceived and designed the experiments, analyzed the data, prepared figures and/or tables, authored or reviewed drafts of the paper, and approved the final draft.
- Shiqing Liu performed the experiments, analyzed the data, prepared figures and/or tables, and approved the final draft.
- Shihong Su and Yang Chen performed the experiments, authored or reviewed drafts of the paper, and approved the final draft.
- Gengyun Sun conceived and designed the experiments, authored or reviewed drafts of the paper, and approved the final draft.

## Human Ethics

The following information was supplied relating to ethical approvals (i.e., approving body and any reference numbers):

The Ethics Committee of the First Affiliated Hospital of Anhui Medical University approved this research (anyiyifuyuanlunshen-Quik-PJ-2019-09-09; anyifuyuan-Quik-PJ-2020-05-08).

## Data Availability

The raw measurements are available in the Supplementary Files.

## Supplemental Information

Supplemental information for this article can be found online at http://dx.doi.org/10.7717/peerj.10470#supplemental-information.

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
