# Peer review of "Construction and validation of a novel prognostic signature of microRNAs in lung adenocarcinoma"

_PeerJ, doi:10.7717/peerj.10470_

## Round 0.1 · original submission · Major Revisions

Three specialists in the field evaluated the present manuscript, and both have concerns related to this paper. The reviewers have described all points that should be answered by the authors. Please ensure that the English language in this submission meets our standards: uses clear and unambiguous text, is grammatically correct, and conforms to professional standards of courtesy and expression. Considering the evaluation carried out by the reviewers, I recommend major revision in this submission.

Reviewer 1 ·

Basic reporting

no comments

Experimental design

no comments

Validity of the findings

no comments

Additional comments

The authors performed an analysis using TCGA data to identify a miRNA signature with prognostic value in lung adenocarcinoma patients. They found a signature of 6 miRNAs associated with prognosis.
A series of points should be clarified:
- The end point of the statistical analysis should be better clarified and specified
- Did the statistical analysis consider the adjuvant or neoadjuvant treament variable? This should be specified
- Figure 4 is related to "gastric cancer" patients. May be there is a mistake
- The english languages should be revised

·

Basic reporting

There are multiple grammatical and stylistic errors over the text. The English should, therefore, be thoroughly revised by a Native speaker (ideally professional proofreader).
Lines 112-113. The phrase “... significant correlation with lung adenocarcinoma” is not clear. Please indicate the exact parameter with which correlation of eight miRNA expression was observed (e.g. stage of the disease, the aggressiveness of the tumor, etc.).
Lines 111-123. From the sentence “The Cox univariate regression analysis was 112 employed for the preliminary screening of potential miRNAs” it remains unclear what is meant after “potential miRNA”. Please clarify.
Figure legends 4, 5 and 6 should include more information (description) of the corresponding plot.

Experimental design

The MMs and results sections are too short and should contain more details. Specifically, the calculations and data analysis procedure should be described in more detail (ideally with exact formulas used) and include references to the corresponding methodology literature or reports where such calculations are described. Next, the format in which the miRNA expression data were collected as well as the method used to measure miRNAs expression should be indicated. Ideally, the raw miRNA data downloaded from the TCGA database should be included in the supplementary material. Finally, it is not described how exactly (methodology, software, etc.) Gene Ontology and KEGG analysis were performed.

Validity of the findings

The reference list must be rechecked and corrected. For instance, reference 5 is incomplete (lacking the authors' names). The reference 26 seems to lack the page numbers. Reference 29 lacks the journal name. References 39 and 43 have no publication year and journal volume/pages indicated. Some references contain the indicated months and the others are not.

Additional comments

In their work entitled “Construction and validation of a novel prognostic signature of microRNAs in lung adenocarcinoma” the authors analyzed miRNA expression data in a large number (499 in total) tumor specimen obtained from Cancer Genome Atlas (TCGA) database. Furthermore, preliminary miRNA screening using Cox univariate regression analysis resulted in a selection of eight miRNAs which demonstrated a statistically significant correlation with disease status; next, using LASSO cox regression analysis the authors established a prognostic signature for overall survival which included miR-1468, miR-299, miR-4709, miR-5571, miR-582 and miR-3653. Finally, the authors showed that indicated target genes of the above-mentioned miRNAs were related to Wnt and mTOR signaling pathways, which are strongly associated with carcinogenesis. The presented miRNA signature has the potential to be used for personalized therapy strategies for lung cancer patients.
All statistical calculations seem to be performed adequately; however, a more detailed description is necessary. The data are convincingly presented in the figures. The article can be reconsidered for a publication only after several points mentioned above are addressed.

Reviewer 3 ·

Basic reporting

The manuscript by Li et al. provides a 6-miRNA signature with prognostic value for overall survival y patients with lung adenocarcinoma. Background/context has been adequately provided and the references are appropiated. The following caveats need to be solved before publication:

1.-English used throught the text needs to be extensively revised. It lacks clarity and is often confusing. Some examples:

Abstract:"Lung cancer (LC) is highly implicated in biological process" (makes no sense)
p. 3, l. 41: "...which is the most common pathological type of LC" (all cancers are pathological)
p.3, l. 46: "The limited survival benefits indicate the deficiency of the current staging system and the presence of more unknown tumor characteristics" (needs proofing and rephrasing)
p. 3, l. 57: "With consistency" change to "Consistently"
p. 4, l. 71: "And it is of promising value..." (no sense) change to "It is promising..."
p. 4, l. 72: "Herein, in our study..." is redundant, just leave "Herein"
p. 4, l. 76: "Material and Method" change to "Material and Methods"
p. 5, l. 86: "...resample model inclusion proportion (RMIP)" change to "...resample model inclusion probability (RMIP)"
Throught the text OS is defined as "Over Survival" instead of "Overall Survival"

These are jus a few correlative examples but it goes on throught the entire manuscript.

2.-Raw data for the Cox univariate regression analysis has not been provided. This is an important flaw. The authors only provide final Kaplan-Meyer graphs for their "selected 8 miRNAs".

3.-The identification of potential target genes remains speculative without proper assessment by immunohistochemistry in tumor samples.

Experimental design

The work presented in this manuscript is original and withing the aims and scope of the jounal. The experimental design suffer from the following deficits:

1.-As already mentioned, this study heavily relays on the statistical analysis of tumor expression data from the TCGA database, however, the raw data obtained after the analysis has not been provided.

2.-Fig. 1A: the Y-axis is labeled as "frecuency", however the scale consist of absolute numbers (samples)

3.-p. 5, l. 121: a risk score formula is provided without any further explanation nor is the methodology to get such formula explained.

4.- p. 5, l. 126: the authors define high risk and low risk groups of patients, however it is not clear which group corresponds to high or low levels of the 6-miRNAs.

5.-p. 6, l. 150: the authors use different well-stablished algorithms to obtain a miRNA-mRNA regulatory network and predict ontology terms and putative targets. However, none of the identified targets (their expression levels) has been assessed in a tumor samples.

The methos section could be improved by extending the description of the different methodologies ( (particularly, the statistics) and being more specific in the details.

Validity of the findings

-The lack of some raw data (as mentioned), together with the above-mentioned flaws in the experimental design and the too-speculative nature (not supported by experimental data) of some predictions regarding genes potentialy regulated by the 6-miRNA signature limit the scientific soundness of the manuscript.

---

## Round 0.2 · Minor Revisions

The Rebuttal document that is provided is very difficult to navigate. It has too much content and is spread over 90 pages. I am requesting you to revise the document:

1. Please reduce page margins and line-spacing so that the document is more compact.

2. Please do not insert figures or tables that are in the manuscript or its SI. Just refer to them in the Rebuttal document (Fig. S1, etc.). The document will be shorter this way.

3. Similarly remove text portions of manuscript that you have provided in the Rebuttal. Just refer to the appropriate section in the manuscript. By looking at the version of the manuscript with changes highlighted, reviewers will be able to know what exactly was changed.

4. Remove any unnecessary styling (text box, color, border, etc.) in the Rebuttal document. They are distracting. Please just use plain format as much as possible. Some bold format or a different color can be used for headings and for reviewer comments that are quoted.

5. Your response should be short but adequate in detail. E.g., if a reviewer has pointed to a spelling mistake and you have corrected it, just state something like: "This mistake was corrected." Another example: "We thank the reviewer for pointing this study. We have now added it as a reference in the Discussion section."

6. Please provided responses in a point-by-point format. E.g.

Reviewer # 1

1. Comment 1
Your response

2. Comment 2
Your response

...

Reviewer #2

...


[

---

## Round 0.3 · Minor Revisions

The revised version has now been examined by one of the three reviewers who had looked at the original submission. This reviewer has a few comments, which are minor. Kindly address these comments with a new version of the manuscript.

·

Basic reporting

The authors adequately responded to the concern raised in the first round of revisions, however, I do have two more issues:
1. The authors provide AUC values both on training and validation groups (section Validation and evaluation of the miRNAs signature). However, they do not provide values of sensitivity and specificity. It is really important to do this since AUC only measures "classification potential" of the classifier ignoring actual accuracy (AUC can be really high with very low accuracy). As we can see, these values actually are presented at the bottom of ROC curves in Figure 2, however, they should be clearly stated in the text.
2. In the Data and sources, section authors state that the LUAD patients were randomly assigned to the training and validation groups. Can the authors repeat this step multiple times to assess what will happen with AUC, sensitivity and specificity? Obviously, other partitions to training and validation groups can both increase and decrease the reliability of the classifier. Thus, it will be helpful to see means/variations/confidence intervals.

Experimental design

The authors provide the original research within aims and scope of the journal.

Validity of the findings

Please see the second issue upper.

---

## Round 0.4 · accepted · Accept

The minor comments raised in the previous round of review have been adequately addressed in this revised version of the manuscript.